# Genome-Wide Identification, Molecular Characterization, and Involvement in Response to Abiotic and Biotic Stresses of the HSP70 Gene Family in Turbot (*Scophthalmus maximus*)

**DOI:** 10.3390/ijms24076025

**Published:** 2023-03-23

**Authors:** Weiwei Zheng, Xiwen Xu, Yadong Chen, Jing Wang, Tingting Zhang, Zechen E, Songlin Chen, Yingjie Liu

**Affiliations:** 1Yellow Sea Fisheries Research Institute, Chinese Academy of Fishery Sciences, Qingdao 266071, China; 2College of Fisheries and Life Science, Shanghai Ocean University, Shanghai 201306, China; 3Shandong Key Laboratory of Marine Fisheries Biotechnology and Genetic Breeding, Qingdao 266071, China; 4Chinese Academy of Fishery Sciences (CAFS), Beijing 100141, China

**Keywords:** *Scophthalmus maximus*, heat shock protein 70 (HSP70), abiotic stress, biotic stress

## Abstract

Heat shock proteins 70 (HSP70s) are known to play essential roles in organisms’ response mechanisms to various environmental stresses. However, no systematic identification and functional analysis has been conducted for HSP70s in the turbot (*Scophthalmus maximus*), a commercially important worldwide flatfish. Herein, 16 HSP70 genes unevenly distributed on nine chromosomes were identified in the turbot at the genome-wide level. Analyses of gene structure, motif composition, and phylogenetic relationships provided valuable data on the HSP70s regarding their evolution, classification, and functional diversity. Expression profiles of the HSP70 genes under five different stresses were investigated by examining multiple RNA-seq datasets. Results showed that 10, 6, 8, 10, and 9 HSP70 genes showed significantly up- or downregulated expression after heat-induced, salinity-induced, and *Enteromyxum scophthalmi*, *Vibrio anguillarum,* and Megalocytivirus infection-induced stress, respectively. Among them, *hsp70* (*hspa1a*), *hspa1b*, and *hspa5* showed significant responses to each kind of induced stress, and qPCR analyses further validated their involvement in comprehensive anti-stress, indicating their involvement in organisms’ anti-stress mechanisms. These findings not only provide new insights into the biological function of HSP70s in turbot adapting to various environmental stresses, but also contribute to the development of molecular-based selective breeding programs for the production of stress-resistant turbot strains in the aquaculture industry.

## 1. Introduction

Heat shock proteins (HSPs), first discovered in the salivary gland cells of the *Drosophila melanogaster* [1], are ubiquitously distributed in the cells of all organisms from bacteria to humans. HSPs are a large class of highly conserved molecular chaperons whose expression is induced by heat or other forms of stress [2]. Some of the HSP family members are known as housekeeping proteins that play distinct roles in the folding, assembling, secretion, intracellular localization, regulation, and degradation of proteins within unstressed cells. HSPs are generally classified into six distinct families, HSP110, HSP90, HSP70, HSP60, HSP40, and small HSPs, based on their molecular weight (MW) [3,4]. They are commonly present in the cytosol, mitochondria, endoplasmic reticulum, and nucleus [5]. Among all those in HSP families, HSP70 genes are the most evolutionarily conserved and have attracted considerable attention from researchers.

HSP70 genes contain three main conserved domains, a 40-kDa N-terminal ATPase domain (NBD) controlling the interaction with the client protein, an 18-kDa substrate-binding domain (SBD) that identifies the hydrophobic regions in a client protein during the initial stages of client protein folding, and a 10-kD variable C-terminal domain [6,7]. Based on their functions and regulatory patterns, HSP70 genes are grouped into two distinct forms: constitutively expressed heat shock proteins (HSCs), which are present in cells under normal physiological conditions, and inducible heat shock proteins, which are expressed in response to specific stress conditions [8]. A substantial amount of evidence has suggested that HSP70 genes not only play crucial roles in promoting protein folding and proteostasis [4,9], the degradation of misfolded proteins [10], the membrane translocation of proteins [5], and the prevention of deleterious protein aggregation [11] and other cell-protective biological processes, but also exhibit essential functions in response to diverse environmental stresses (such as heat [12], hypoxia [13], pathogen infection [14], and heavy metal and organochlorine exposure [15]), physiological stress [16], and the immune response [17]. In summary, HSP70 genes are indispensable for maintaining protein homeostasis and cellular recovery after exposure to various environmental stresses.

Fish are generally exposed to various stresses in the water environment, such as cold or hot temperatures, hypoxia, low- or high-salinity stress, and other challenges caused by pathogen infection. Given the crucial functions of HSP70 genes in response to environmental stresses coupled with the great abundance of available genome sequence resources, an increasing number of HSP70 gene families have been identified at the genome-wide level in multiple teleosts, and the potential roles of these genes in response to abiotic and biotic stresses have also been described in recent years. For instance, 17 HSP70 genes were systematically characterized and analyzed in the spotted seabass (*Lateolabrax maculatus*); among them, 4, 11, and 7 gene family members were found to be involved in the response to heat, hypoxia, and alkalinity stress, respectively [18]. In the large yellow croaker (*Larimichthys crocea*), 17 HSP70 genes were identified at the genome-wide level; moreover, 6 gene family members were significantly up- or downregulated in liver tissue after exposure to cold and heat stresses, as determined with transcriptome data [19]. Moreover, a total of 20 HSP70 gene family members have been found in the *Boleophthalmus pectinirostris*, with most of these genes downregulated after exposure to high concentrations of ammonia [20]. Furthermore, 14 HSP70 genes have been detected in the *Cynoglossus semilaevis* through database searches and by manual selection, and expression profiles have demonstrated that most of these gene family members, especially *hspa5,* showed upregulated expression after exposure to low-salinity stress [21]. Moreover, a total of 16 HSP70 genes were detected in the genome of the rainbow trout (*Oncorhynchus mykiss*); in addition, an RNA-seq analysis indicated that 4 and 7 HSP70 gene family members showed significantly upregulated expression after heat stress in liver and head kidney tissues, respectively [22]. In the channel catfish (*Ictalurus punctatus*), 16 HSP70 genes were detected via a combination of genome databases and RNA-seq data, and 9 and 2 gene family members showed significant expression in gill tissue after a challenge with the *Flavobacterium columnare* and in intestine tissue after infection with the *Edwardsiella ictaluri* [14]. In the Japanese flounder (*Paralichthys olivaceus*), 15 HSP70 genes were identified, among which 5 gene family members were highly expressed post-infection with *Edwardsiella tarda* [23]. All of the above studies demonstrated that HSP70 genes play potentially important roles in the biological mechanisms underlying the responses to biotic and abiotic stresses in teleosts. However, no systematic identification or functional analysis of the HSP70 gene family has been performed in the turbot to date. 

The turbot (*Scophthalmus maximus*, FishBase ID: 1348) is an economically important cold-water flatfish species native to the Northeast Atlantic [24]. Due to its tasty meat and high nutritional value, the turbot has become the marine flatfish produced at the highest level under aquaculture conditions worldwide [25]. Unfortunately, due to high-density industrial farming coupled with an increased frequency in extreme weather events, in recent years, the turbot has faced a variety of abiotic and biotic stresses during aquaculture, such as heat stress induced by global warming [26], salinity stress caused by disrupted climate conditions and tidal water flow [27,28], as well as pathogen-induced stresses resulting from high-density industrial farming [29,30,31,32]. These environmental stresses severely threatened the welfare or caused elevated mortality in this species under aquaculture conditions, leading to huge economic losses that greatly impede the healthy and sustainable development of the turbot industry. Because of the essential functions of HSP70 genes in response to diverse stresses, systematically identifying and exploring their expression patterns under abiotic and biotic stresses is expected to be valuable for further functional studies and exhibit important practical significance in the development of efficient farming practices, thereby leading to an increase in turbot fitness in aquaculture production. Fortunately, high-quality genome assembly [33] and multiple published RNA-seq datasets resources [26,27,28,30,31,32,34,35] made available in recent years make it possible to perform systematic identification, phylogenetic analysis, and a functional study of HSP70 genes in the turbot. 

In the present study, a complete set of HSP70 genes in the turbot was first identified at the genome-wide level. Molecular characterizations, including biochemical properties, protein structure, and molecular evolution, were also determined. Furthermore, evolutionary relationship, exon–intron structure, and conserved domain and motif analyses were carried out, respectively. In addition, to better understand the role of HSP70 genes challenged by a variety of environmental stresses in the turbot, the gene expression patterns in distinct tissues after exposure to diverse abiotic and biotic stresses, such as heat, salinity, and three different pathogen infection stresses, were investigated by analyzing multiple stress-related RNA-seq datasets. Finally, qPCR analyses were performed to further validate the functions of the HSP70 gene candidates that showed comprehensive anti-stress responses to both abiotic and biotic stresses, as determined by the results of RNA-seq data analyses. This may be the first systematic identification and functional investigation of HSP70 genes in the turbot. The findings of this study will aid in the study of the molecular regulatory mechanism of HSP70 genes in response to environmental stresses and provide new insight into increasing the environmental adaptation and comprehensive stress resistance of the turbot. 

## 2. Results

### 2.1. Identification and Characteristics of HSP70 Genes in Turbot

In the present study, a total of 16 HSP70 genes were identified in the turbot at the genome-wide level using a combination of BLASTp and HMM searches and confirmation with an NCBI CDD scan. Then, turbot HSP70 genes were named following the nomenclature of *D. rerio* heat shock proteins. Information on the domains of turbot HSP70 genes is shown in Table 1 and Appendix A. Comparisons of the numbers of HSP70 genes in humans and other well-characterized teleosts are shown in Appendix A, indicating their high evolutionary conservation among species. In addition, the basic physicochemical properties of proteins encoded by HSP70 genes in the turbot are shown in Table 1 and Appendix A. Specifically, the number of amino acids varied from 431 (*hspa8a.1*) to 986 (*hyou1*). The predicted molecular weights (MWs) ranged from 48.13 kDa (*hspa8a.1*) to 110.91 kDa (*hyou1*), and the theoretical isoelectric point (pI) ranged from 5.00 (*hspa5*) to 8.48 (*hspa12b*). Furthermore, *hspa12b* showed the longest conserved domain at 497 amino acids, whereas the shortest domain was found in *hspa8a.1* (315 amino acids). Moreover, the grand average of hydropathicity (GRAVY) values of turbot HSP70 proteins were all negative, indicating that all the proteins were hydrophobic. Moreover, the prediction of the protein instability index (II) (a reference value for protein stability in vitro analysis) showed that 7 of the 16 HSP70 proteins were identified as stable proteins (II < 40), with the others predicted to be unstable (II > 40). Investigation of the molecular properties of HSP70 proteins aids in determining their biological functions. 

### 2.2. Phylogenetic Analysis of HSP70 Genes among Teleost Species 

To clarify the phylogenetic relationships and the classification of HSP70 genes among different teleost species, an unrooted NJ phylogenetic tree was constructed using HSP70 protein sequences from the turbot and three other representative teleost species (Appendix A). As shown in Figure 1, 66 HSP70 genes were classified into nine distinct groups, the hspa1, hspa4, hspa5, hspa8, hspa9, hspa12, hspa13, hspa14, and hyou1 groups. All of the turbot HSP70 genes were classified into these nine different clades, and they were first grouped with orthologous genes of selected species with strong bootstrap values. In addition, two genes (*hspa1b* and *hsp70*), two genes (*hspa12a* and *hspa12b*), three genes (*hspa4a*, *hspa4b*, and *hspa4l*), and four genes (*hspa8a.1*, *hspa8a.2*, *hspa8b*, and *hsc70*) were identified as duplicated turbot HSP70 genes in the hspa1, hspa12, hspa4, and hspa8 groups, respectively. In contrast, no gene duplications were found in the hspa5, hspa9, hspa13, hspa14, or hyou1 groups, which was similar to the results obtained in an analysis of three other teleost species. 

### 2.3. Chromosomal Distribution and Selection Test of Duplicated HSP70 Genes 

Sixteen HSP70 genes were unevenly distributed on nine turbot chromosomes (Figure 2). Specifically, six HSP70 genes, *hspa8a.1*, *hspa8a.2*, *hspa4a*, *hspa12b*, *hspa14,* and *hyou1*, were located on Chr2. In addition, two HSP70 genes were observed on Chr 9 and Chr 11. In addition, the remaining six HSP70 genes were distributed on different chromosomes, such as Chr 1, Chr 3, Chr 10, Chr 15, Chr 17, and Chr 20. In terms of duplicated genes, *hspa8a.1* and *hspa8a.2* were located on the same chromosome, while the other duplicated gene pairs were located on different chromosomes. To investigate whether these duplicated HSP70 genes experienced various selection pressures, the nonsynonymous (*Ka*), synonymous (*Ks*), and *Ka*/*Ks* ratios for all duplicated HSP70 gene pairs were calculated (Table 2). The results showed that the *Ka*/*Ks* ratios of all the duplicated HSP70 gene pairs ranged from 0.04777 to 0.2558, which were much lower than 1.0, indicating that they had all been subjected to purifying selection.

### 2.4. Exon–Intron Structure and Motif Analysis of HSP70 Genes

To better clarify the evolutionary conservation and functional diversification of the HSP70 gene family, exon–intron constitution and motif analyses were carried out. As shown in Figure 3, the number of exons in HSP70 genes varied greatly from 2 to 25. However, when these data were combined with phylogenetic analysis data, paralogous genes derived from the same group were shown to possess similar gene structures. For instance, both *hsp70* (*hspa1a*) and *hspa1b* in group hspa1 contained only one intron and two exons, which was a composition obviously different than the members in the other groups. In addition, a total of 21 putative motifs were detected in HSP70 genes, as shown in Figure 4. On the whole, HSP70 genes from the same group exhibited similar motif patterns; for instance, paralogous genes in the hspa4 group (*hspa4a*, *hspa4b*, and *hspa4l*), hspa12 group (*hspa12a* and *hspa12b*), and hspa1 group (*hspa1b* and *hsp70* (*hspa1a*)) shared identical motif patterns. Moreover, most HSP70 genes shared eight conserved motifs, motifs 1, 2, 4, 5, 7, 8, 9, and 12. Moreover, the number of motifs in HSP70 genes in different groups ranged greatly from 2 to 21. The results of gene structure and motif analyses provided potent support for the phylogenetic classification and functional diversification reported for the HSP70 gene family.

### 2.5. Subcellular Localization and Protein Structure Analysis

The prediction of subcellular localization indicated that most HSP70 proteins in the turbot were located in the cytosol and extracellular space; however, the proteins encoded by *hspa9*, *hspa12b*, and *hyou1* were localized in the mitochondria, nucleus, and endoplasmic reticulum, respectively (Table 3). Moreover, the secondary structure prediction showed that turbot HSP70 proteins were primarily composed of alpha helices (32.39–50.51%) and random coils (29.61–44.48%), while extended strands and beta turns accounted for 12.37–23.72% and 3.31–7.37% of the structure, respectively (Table 3). In addition, the signal peptide prediction results demonstrated that only hyou1 and hspa5 encoded proteins with Sec/SPI signal peptides with SP values of 0.995566 and 0.999579, respectively, and these gene products were preliminarily identified as secretory proteins. As shown in Figure 5, the cleavage site was between positions 27 and 28 with a marginal probability of 0.636974 in *hyou1*, while in *hspa5* the cleavage site was between positions 16 and 17 with a marginal probability of 0.982612. 

### 2.6. Expression Patterns of HSP70 Genes under Abiotic Stresses

To determine the involvement of the turbot HSP70 genes in response to abiotic stress, the expression profiles of HSP70 genes were investigated utilizing heat-stress-related and low- and high-salinity-stress-related RNA-seq datasets (Figure 6). Under heat stress, the expression patterns of HSP70 genes in the kidney and liver tissues were detected (Figure 6A). The detailed expression data are provided in Appendix A. In the kidney, a total of 10 genes, *hsc70*, *hsp70* (*hspa1a*), *hspa1b*, *hspa4a*, *hspa4l*, *hspa5*, *hspa9*, *hspa8a.1*, *hspa8a.2,* and *hyou1*, showed differential expression after heat treatment compared to the control group. By comparison, only six genes, *hsp70* (*hspa1a*), *hspa1b*, *hspa4a*, *hspa4l*, *hspa5,* and *hyou1*, were differentially expressed in the liver tissue after heat treatment. The expressions of almost all of these differentially expressed (DE) HSP70 genes in the kidney and liver tissues were significantly continuously upregulated as the temperature increased. In addition, in both liver and kidney tissues, the number of DE HSP70 genes was found to be the highest in the 30 ℃ (T3) group compared with the other heat treatment groups (T1 and T2). 

Under low- and high-salinity stresses, the expression profiles of HSP70 genes in the kidney, gill, and liver tissues were illustrated (Figure 6B). The detailed expression data are provided in Appendix A. On the whole, HSP70 genes exhibited obviously distinct expression levels in the kidney, gill, and liver tissues. In the gills, six DE HSP70 genes were detected in the low- and high-salinity groups compared to the control group (natural seawater); among these genes, *hspa5* and *hspa8a.1* showed significantly up- and downregulated expression under low-salinity (5‰) stress conditions, respectively. Under high-salinity (50‰) stress conditions, *hsp70* (*hspa1a*), *hspa12a,* and *hspa1b* showed significantly upregulated expression; in contrast, the expression of *hspa12b* was significantly downregulated. However, no DE HSP70 genes were identified in the kidney or liver tissues. 

In a word, four HSP70 genes, *hsp70* (*hspa1a*), *hspa1b*, *hspa5*, and *hspa8a.1*, were significantly induced both by heat and salinity stresses, indicating their essential roles in response mechanisms to various abiotic stresses in turbot.

### 2.7. Expression Patterns of HSP70 Genes under Biotic Stresses

Using pathogen (parasite, bacteria, and virus)-challenged RNA-Seq datasets, the expression profiles of turbot HSP70 genes in different tissues under biotic stress were determined (Figure 7). The detailed expression data are provided in Appendix A. After parasite (*E. scophthalmi*) infection, eight DE HSP70 genes (*hsp70* (*hspa1a*), *hspa4a*, *hsc70*, *hspa1b*, *hspa9*, *hspa8b*, *hspa4l*, and *hspa5*) identified in four distinct tissues showed different expression than in the healthy group. Specifically, in the kidney, *hsp70* (*hspa1a*) and *hspa4a* were upregulated both at 24 and 42 days post-infection (dpi) with *E. scophthalmi*. In the pyloric caeca, *hsc70*, *hsp70* (*hspa1a*), *hspa1b*, *hspa4a*, and *hspa9* were all upregulated at 42 dpi, while no DE HSP70 genes were found at 24 dpi. In addition, no DE HSP70 genes were detected in the spleen. In the thymus, only the expression of *hspa8b* was detected significantly upregulated at 42 dpi. Moreover, in the blood with four different levels (incipient, slight, moderate. and severe) of *E. scophthalmi* infection, DE HSP70 genes, including *hspa4l* and *hspa5*, were found only in the severely infected group, and they all had upregulated expression.

Following bacterial (*V. anguillarum*) challenge, 10 DE HSP70 genes, *hsp70* (*hspa1a*), *hspa13*, *hspa1b*, *hspa4a*, *hspa4b*, *hspa4l*, *hspa5*, *hspa9*, *hyou1*, and *hspa12b*, were detected in five different tissues compared to the control group. Specifically, in the intestine, three DE HSP70 genes, *hsp70* (*hspa1a*), *hspa4a,* and *hspa5*, were all upregulated at 1 h after challenging with *V. anguillarum* and then were downregulated continuously at 4 h and 12 h. In the spleen, no DE HSP70 genes were identified. In addition, *hsp70* (*hspa1a*), *hspa1b*, and *hspa12b* were differentially expressed in the gill, and among these genes, the expression of *hsp70* (*hspa1a*) and *hspa1b* was upregulated, while that of *hspa12b* was downregulated. Furthermore, in the kidney, two DE HSP70 genes, *hsp70* (*hspa1a*) and *hspa4l*, had significantly upregulated expression. Moreover, seven DE HSP70 genes, *hspa4a*, *hyou1*, *hspa5*, *hspa13*, *hspa9*, *hspa4b,* and *hsp70* (*hspa1a*), had upregulated expression in the liver. 

After infection with a virus (*Megalocytivirus*), a total of nine DE HSP70 genes, *hsp70* (*hspa1a*), *hspa12b*, *hspa1b*, *hspa4a*, *hspa4b*, *hspa4l*, *hspa5*, *hspa9,* and *hyou1*, were detected in the head kidney. In detail, at 3 dpi, no DE HSP70 genes were found. By comparison, *hsp70* (*hspa1a*), *hspa1b*, *hspa5*, *hspa4a*, *hyou1*, *hspa4l*, and *hspa12b* were differentially expressed at 6 dpi, and they were all upregulated except for *hspa12b*, which showed downregulated expression. In addition, the DE HSP70 genes identified at 9 dpi, including *hsp70* (*hspa1a*), *hspa4b*, *hspa9*, *hspa4l*, *hyou1*, *hspa4a*, *hspa5*, and *hspa1b*, all had upregulated expression. 

In summary, a total of six HSP70 genes, *hsp70* (*hspa1a*), *hspa1b*, *hspa4a*, *hspa4l*, *hspa5*, and *hspa9*, were significantly affected by three different pathogens, demonstrating their essential roles in response to various biotic stresses.

### 2.8. qPCR Validation of HSP70 Gene Candidates Involved in Comprehensive Anti-Stress Responses

Based on the expression pattern analysis of HSP70 genes under both biotic and abiotic stresses determined using RNA-seq datasets, we concluded that *hsp70* (*hspa1a*), *hspa1b*, and *hspa5* may be the HSP70 gene family members with comprehensive anti-stress functions. To further validate their potential comprehensive anti-stress functions, qPCR analyses were conducted. We first validated the expression patterns of *hsp70* (*hspa1a*), *hspa1b*, and *hspa5* in the liver and kidney tissues of the turbot at 24 h after heat stress (Figure 8A,B). The results showed that *hsp70* (*hspa1a*), *hspa1b*, and *hspa5* exhibited similar expression patterns, that is, an extremely significant upregulation in expression continuously as the temperature increased in the liver and kidney tissues after heat stress. This finding was consistent with the results of the RNA-seq analysis, indicating that *hsp70* (*hspa1a*), *hspa1b*, and *hspa5* do play important roles in response to abiotic stress, especially heat stress. We then also systematically illustrated the expression patterns of *hsp70* (*hspa1a*), *hspa1b*, and *hspa5* in the kidney and spleen tissues of the turbot at 0, 6, 12, 24, and 48 h after infection with *E. tarda* using qPCR (Figure 9). The results showed that *hsp70* (*hspa1a*), *hspa1b*, and *hspa5* were all significantly or extremely significantly upregulated in the spleen and kidney tissues at 6, 12, 24, and 48 h post-infection with *E. tarda*, especially *hsp70* (*hspa1a*) and *hspa1b*, indicating their crucial roles in response to biotic stress.

## 3. Discussion

Heat shock protein 70 (HSP70) gene family members are involved in various housekeeping and stress-related activities in all organisms from bacteria to mammals [36,37]. Due to their important roles, especially in enhancing the response of cells to various stress stimuli, the HSP70 gene family has been extensively studied in a variety of aquaculture species, including invertebrate [38], bivalve [39], and multiple teleost species [14,18,19,20,21,23]. Nevertheless, a systematic analysis of HSP70 genes in the turbot has been lacking. In the present study, we conducted comprehensive research on the HSP70 gene family in the turbot, including the systematic identification of genes at the genome-wide level and analysis of biochemical characteristics, phylogeny, gene structure, and conserved motifs. In addition, to determine the involvement of these genes in mechanisms associated with responses to abiotic and biotic stresses, the expression patterns of HSP70 genes in the turbot under heat-induced, salinity-induced, and parasitic, bacterial, and virus infection-induced stresses were determined by examining multiple RNA-seq datasets. Importantly, qPCR was performed to further verify the potential functions of the HSP70 gene candidates showing comprehensive anti-stress in response to both biotic and abiotic stresses.

In the present study, a total of 16 HSP70 genes were detected in the turbot, which was more than the number previously found in crustaceans (*Portunus trituberculatus*) (9) [40] but considerably less than the number found in bivalve species, such as the *Crassostrea gigas* (88) [41], *Mercenaria mercenaria* (133) [39], and *Cyclina sinensis* (60) [39], and similar to the number in humans (17) [42] and in other teleost species (14-20). Although the number of HSP70 genes varies among species, they display high-level evolutionary conservation in humans and teleost species. The differences in gene number might be a result of gene duplication or gene loss during the process of evolution. In addition, the turbot contained almost all the orthologs of HSP70 genes in zebrafish and other teleosts, with the exception of *hsph1*, for which an ortholog has been found only in zebrafish, not other teleost species. We speculated that *hsph1* might have been lost from the genome of most teleosts during evolution. Furthermore, the phylogenetic analysis of four representative teleost species showed that the HSP70 genes within the same group were more similar to each other than to those in other groups, demonstrating that HSP70 genes were highly conserved during teleost evolution. 

Gene duplication represents the principal mechanism by which gene families expand and provides raw materials for the evolution of novel gene functions [43,44,45]. In this study, we observed that duplicated HSP70 genes were mainly contained in the hapa1, 4, 8, and 12 groups in all teleosts. In the turbot, the genes *hspa8a.1* and *hspa8a.2* located on Chr 2 were identified in tandem duplications [46], whereas other genes were distributed on different chromosomes, indicating that whole-genome duplication and tandem duplication events both contributed to the evolution and expansion of HSP70 genes in the turbot. Classic examples of gene family expansion have been found in bivalves; for example, 88 HSP70 gene copies have been identified in the *C. gigas* [41], and 133 copies have been identified in the *M. mercenaria* [39]. Studies on bivalves have demonstrated that the expansion of HSP70 genes was probably crucial to the environmental adaptation process in the highly stressful intertidal zone [39,41]. In our study, duplicated genes, that is, *hsp70* (*hspa1a*) and *hspa1b*, were differentially expressed under five stress conditions (heat-, salinity-, and parasitic, bacterial, and virus infection-related stress conditions). Moreover, the expansion of *hspa1* might have contributed to the terrestrial adaptations in *B. pectinirostris*, which enabled them to spend a considerable part of their lives on land, during evolution [20]. In addition, the *Ka*/*Ks* ratios of all repetitive HSP70 gene pairs in the turbot were significantly less than 1 (varying from 0.047773 to 0.255827), indicating that repeated genes in the turbot underwent a strong purifying selection during evolution.

Gene structure provides strong support for evolutionary classifications within a gene family [47]. Herein, a combination of the results from phylogenetic analysis and gene structure analysis showed that paralogous gene pairs, that is, HSP70 genes derived from the same group, showed highly similar exon–intron distribution patterns and a conserved motif constitution, demonstrating their high evolutionary conservation and similar functions within a group. In addition, numerous studies have proven that distinct exon–intron structures and motif patterns in HSP70 genes are related to their unique biological functions [19,48,49]. In the turbot, the numbers of exons (ranging from 2 to 25) and motifs (ranging from 2 to 21) varied considerably between groups, indicating their potential biological function diversification among members in diverse groups. Interestingly, *hsp70* (*hspa1a*) and *hspa1b*, each with only one intron and two exons, were significantly induced by all five evaluated stress conditions, which might have been due to the low number of introns and exons, allowing efficient transcription. 

To detect the involvement of HSP70 genes in abiotic stress, the expression profiles of HSP70 genes after turbot were challenged by heat and salinity stresses were illustrated for the first time. Under heat stress treatment, 10 and 6 DE HSP70 genes were detected in the kidney and liver, respectively. With increasing temperature, almost all of these genes had significantly and continuously upregulated expression, and the number of DE HSP70 genes was clearly increased. Similar results have also been found in the *L. maculatus* and *O. mykiss*, in which most of the examined HSP70 genes showed significantly upregulated expression in different tissues after exposure to heat stress [18,22]. Moreover, four significantly upregulated HSP70 genes in both kidney and liver tissues were identified, *hsp70* (*hspa1a*), *hspa1b*, *hspa5*, and *hspa8a.1*. Among these genes, *hsp70* (*hspa1a*) was one of the most intense responsive genes in the liver, muscle, and gill tissues after heat stress in the spotted seabass [18]. Furthermore, the expression of *hsp70* (*hspa1a)*, *hspa5*, and *hspa8a* was also significantly increased both in the liver and head kidney of the rainbow trout after heat treatment (24 °C) [22] demonstrating their essential roles in heat stress responses, indicating that these genes might be selected as biomarkers of heat stress. In contrast to the highly inducible expression of more than one half of the HSP70 genes expressed after exposure to heat stress, only four HSP70 genes showed significantly upregulated expression in all tested tissues after exposure to salinity stress; in contrast, the expression of two genes was significantly downregulated. The different expression patterns of HSP70 genes suggested that distinct response mechanisms might underlie HSP70-gene-related responses induced by specific environmental stressors. Similar results have also been reported in the spotted seabass. In analyses to measure the sensitivity of HSP70 genes in response to various environmental stresses, most of these genes exhibited highly inducible expression after heat stress; however, a large proportion of HSP70 genes showed significantly downregulated expression after hypoxia and alkalinity stress challenges [18]. In addition, two and four DE HSP70s were detected under low- and high-salinity conditions, respectively, in gills, demonstrating that HSP70 genes may have been more sensitive to high-salinity stress than to low-salinity stress. Furthermore, no DE HSP70 genes were found in the kidney or liver under either low- or high-salinity stress conditions, demonstrating obvious tissue-specific expression patterns of HSP70 genes under salinity stress conditions. Finally, we concluded that the expression of four HSP70 genes, *hsp70*, *hspa1b*, *hspa5*, and *hspa8a.1*, was significantly induced by both heat and salinity stress exposure, indicating their essential roles in participating in various abiotic stresses. 

To determine the functions of HSP70 genes in participating in biotic stress, we investigated the expression patterns of HSP70 genes in the turbot after challenges with three different pathogens for the first time. The expression of 8, 10, and 9 DE HSP70 genes was detected after parasitic (*E. scophthalmi*), bacterial (*V. anguillarum*), and viral (*Megalocytivirus*) infection, respectively, and most of these genes showed significantly upregulated expression. Among these pathogen-activated genes, six genes (*hsp70*, *hspa1b*, *hspa4a*, *hspa4l*, *hspa5*, and *hspa9*) were differentially expressed under all three diverse pathogen infection stresses, suggesting their involvement in response to distinct biotic stresses. Nevertheless, different expression patterns of HSP70 genes were identified following infection with different pathogens. For example, DE *hsc70* were only detected after a challenge with *E. scophthalmi*, while DE *hspa13* was only identified after *V. anguillarum* infection. Otherwise, different numbers of DE HSP70 genes were also detected in the same tissue of turbot infected with the three different pathogens, that is, two, two, and nine DE HSP70 genes were identified in the kidney after *E. scophthalmi*, *V. anguillarum*, and *Megalocytivirus* infection, respectively. In the channel catfish, different expression patterns of HSP70 genes have also been detected after infection with *Flavobacterium columnare* and *Edwardsiella ictaluri* [14]. In addition, tissue-specific expression patterns have also been observed after these three pathogen infections; for instance, after *E. scophthalmi* infection, two, five, one, and two significantly upregulated HSP70 genes were identified in the kidney, pyloric caeca, thymus, and blood, respectively, whereas no DE HSP70 genes were found in the spleen. Similarly, three, three, two, and seven HSP70 genes showed significantly up- or downregulated expression in the intestine, gill, kidney, and liver after infection with *V. anguillarum*, but no DE HSP70 genes were identified in the spleen. These results suggest that both pathogenesis-specific and tissue-specific expression patterns of HSP70 genes existed in the turbot.

## 4. Materials and Methods

### 4.1. Genome-Wide Identification and Sequence Analysis of HSP70 Genes in Turbot

To identify the full set of HSP70 genes in the turbot, the genome sequence, annotation file, and protein sequences of the turbot were first collected from NCBI databases (GCA_022379125.1) [33]. Then, two strategies were used to search the turbot genome to identify HSP70 genes. First, HSP70 protein sequences of the *Danio rerio* were used as a query database to search against turbot amino acid sequences using the BLASTp programs with default parameters. Second, a hidden Markov model (HMM) search using the HMM profile of the HSP70 domain (PF00012) derived from the Pfam protein family database and the HMM profiles of HSP12a and HSP12b (PTHR14187:SF46 and PTHR14187:SF39) retrieved from the PANTHER classification system as queries was conducted to identify candidate HSP70 proteins in the turbot genome using HMMER software (https://www.ebi.ac.uk/Tools/hmmer/search/hmmsearch, accessed on 1 August 2022) with an *E*-value of 10^−4^. Next, the candidate protein sequences were submitted to the Conserved Domains Database (CDD) of NCBI to further confirm the presence of the conserved domains with an *E*-value of 10^−4^. Next, the physicochemical properties of HSP70 proteins, including the MWs, number of amino acids, the theoretical isoelectric point (pI), GRAVY, and the II, were analyzed using the online ExPASy ProtParam tool. 

### 4.2. Phylogenetic Analysis of the HSP70 Genes

To clarify the phylogenetic relationships and classification of HSP70 genes among different teleost species, the amino acid sequences of HSP70 genes from the turbot and other selected teleost species, such as the *D. rerio*, *Oryzias latipes*, and *Lepisosteus oculatus*, were employed to construct a phylogenetic tree. These protein sequences were downloaded from the NCBI and UniProt databases. Multiple protein sequence alignments were performed using the ClustalW method with default parameters in MEGA 11.0 [50]. Then, a neighbor-joining (NJ) phylogenetic tree was constructed by MEGA 11.0 software with the pairwise-deletion option based on the Jones–Taylor–Thornton (JTT) amino acid substitution model. A phylogenetic analysis was performed using the bootstrap method with 1000 replications to evaluate the support for the predicted phylogenetic relationships. Finally, EvolView (v2.0) (https://www.evolgenius.info/evolview/, accessed on 3 August 2022) was used to visualize the phylogenetic tree.

### 4.3. Chromosome Distribution and Molecular Evolution Analysis

To clarify the chromosomal distribution of HSP70 genes in the turbot genome, information on chromosome length and gene position was obtained from the genome Generic Feature Format (GFF) file, and then the genes were visualized using the gene location visualize function in TBtools (v1.098761) [51]. In addition, to determine the evolutionary constraints and selection pressure on HSP70 genes, the *Ka*, *Ks,* and *Ka*/*Ks* ratio of duplicated HSP70 gene pairs were calculated by the simple *Ka*/*Ks* calculator functions in TBtools. 

### 4.4. Gene Structure Analysis and Conserved Motif Identification

The combination of the phylogenetic relationship and exon–intron structure of turbot HSP70 genes was visualized using the Gene Structure Display Server (GSDS, version 2.0, http://gsds.cbi.pku.edu.cn, accessed on 6 August 2022) with the genome GTF file and the phylogenetic tree file constructed by MEGA 11.0 as the input data. Furthermore, conserved motif identification of the HSP70 proteins was performed using the online program MEME (5.4.1) [52] with the number of motifs set to 21 and the other parameters set to default.

### 4.5. Subcellular Localization and Protein Structure Prediction of HSP70

The subcellular localization of HSP70 proteins was detected by the online WoLF PSORT tool (https://www.genscript.com/wolf-psort.html, accessed on 8 August 2022). Then, the secondary structure of HSP70 proteins was predicted using online SOPMA software (http://npsa-pbil.ibcp.fr/cgi-bin/npsa_automat.pl?page=npsa_sopma.html, accessed on 8 August 2022). Signal peptides were predicted using the online SignalP 6.0 software (https://services.healthtech.dtu.dk/service.php?SignalP, accessed on 8 August 2022). 

### 4.6. RNA-seq Datasets Used for the Expression Profiles of HSP70 Genes under Abiotic Stresses

Five published RNA-seq datasets related to abiotic stress (heat and salinity stress) (Appendix A) were obtained from the NCBI SRA database to investigate the expression profiles of the HSP70 genes in the turbot under abiotic stresses. Specifically, two RNA-seq datasets were used for the expression analysis of the HSP70 genes after heat treatment. In the SRP152627 experiment, turbot individuals were exposed to heat stress (23 °C, 25 °C, and 28 °C). After heat treatment for 24 h, the kidney tissues in the turbot in the heat-treated groups and control group, which were maintained at 14 °C, were collected for RNA-seq [26]. In the SRA study of SRP273870, the liver tissues in the treatment groups exposed to heat stress (20 °C, 24 °C, and 28 °C) for 24 h and in the control group (14°C) were used for RNA-seq [53]. For an expression analysis under salinity stress conditions, three salinity-stress-related RNA-seq datasets were collected. In the SRA study of SRP238143, the gill tissues of turbot after low-salinity (5‰), natural seawater (30‰), and high-salinity (50‰) treatments for 24 h were obtained for RNA-seq [28]. In SRP153594, the kidney tissues of turbot after different salinity (5‰, 30‰, and 50‰) treatments for 24 h were collected for RNA-seq. In SRP277001, the liver tissues of turbot subjected to freshwater (0‰) and natural seawater (30‰) were collected at 24 h and used for RNA-seq [27]. 

### 4.7. RNA-seq Datasets Used for the Expression Profiles of HSP70 Genes under Biotic Stresses

To investigate the expression patterns of the HSP70 genes in the turbot under biotic stress conditions, multiple pathogen-stress-related RNA-seq datasets after parasite (*Enteromyxum scophthalmi*), bacterial (*Vibrio anguillarum*), and viral (Megalocytivirus) challenges were retrieved from the NCBI SRA database (Appendix A). For parasite infection stress, four RNA-seq datasets from 5 distinct tissues in healthy and *E. scophthalmi*-infected turbot were downloaded. In experiments SRP065375 and SRP050607, head kidney, spleen, and pyloric caeca tissues were collected for RNA-seq after challenges with *E. scophthalmi* for 24 and 42 days [31,34]. In the SRA study of SRP308109, blood samples collected from four levels of *E. scophthalmi*-infected (incipient, slight, moderate, and severe) turbot and from healthy turbot were used for RNA-seq [29]. In SRP255305, thymus tissues from healthy and infected turbot at 24 and 42 days post-infection with *E. scophthalmi* were selected for RNA-seq [35]. For bacterial challenge stress, data from five SRA studies were collected. In the SRA study of SRP191266, intestine tissues sampled at 0 h, 1 h, 4 h, and 12 h after infection with *V. anguillarum* were used for RNA-seq [30]. In SRA studies of SRP336094, SRP335896, SRP320422, and SRP319434, the liver, head kidney, gill and spleen tissues of turbot after a *V. anguillarum* challenge were collected for RNA-seq [32]. For viral infection, the SRA data from the SRP347383 study were downloaded. Specifically, head kidney tissues from healthy and infected turbot at 3, 6, and 9 days after infection with *Megalocytivirus* were collected for RNA-seq [54].

### 4.8. Expression Analyses of HSP70 Genes 

To illustrate the expression patterns of HSP70 genes under biotic and abiotic stresses, the reads of the aforementioned stress-related RNA-seq datasets were first aligned to the turbot genome by STAR [55] with default parameters. Next, the featureCounts [56] software program in Subread (v2.0.3) [57] was used to count the reads mapped to the genome to generate read count matrixes. With read count matrixes as inputs, edgeR [58] was used to identify differentially expressed (DE) (or significantly up- or downregulated) HSP70 genes based on the criteria of |log_2_fold change (FC)| > 1 and false discovery rate (FDR) < 0.01. Moreover, transcripts per million (TPM) were calculated based on the read count matrixes. Finally, heatmaps showing the expression patterns of HSP70 genes were generated using TBtools (v1.098761) [51] with normalized TPM data (log_2_ (TPM + 1)).

### 4.9. Heat Stress and Edwardsiella Tarda Challenge Experiment

In the heat stress experiment, 80 turbot individuals (26 ± 2.02 g) were randomly divided into four groups and acclimated at 18 °C for 7 days. Then, the water temperature was increased at a constant rate of 1 °C/h until it reached the desired experimental temperature (22 °C (T1), 26 °C (T2), and 30 °C (T3), and the control group (C) was maintained at 18 °C). After heat stress exposure for 24 h, three individuals per group were randomly sampled. Sampled fish were anesthetized with clove oil, and the liver and kidney tissues were collected and flash-frozen in liquid nitrogen and stored at −80 °C until RNA extraction.

In the *E. tarda* challenge experiment, 60 healthy turbot individuals (25 ± 2.45 g) were randomly divided into two groups, the control group and the *E. tarda* challenge group. *E. tarda* was cultured in lysogeny broth (LB) at 28 °C overnight for 24 h using a shaker incubator. In the challenge group, the fish were challenged by intraperitoneal injection with 100 µL of an *E. tarda* suspension consisting of 10^7^ CFU/mL per 1 g body weight, while the fish in the control group were injected with the same volume of 1x PBS solution. At 0, 6, 12, 24, and 48 h post-injection (hpi), three individuals per group were randomly selected for kidney and spleen tissue collection. The sampling method was the same as that performed in the heat stress experiment.

### 4.10. Quantitative Real-Time PCR (qPCR) 

To further validate the gene expression of the candidate HSP70 genes that responded to both biotic and abiotic stresses, qPCR was performed. The primers used to amplify the genes and *β-actin* (the internal control) were designed using Primer-BLAST of NCBI, and the sequences are listed in Appendix A. qPCR assays were performed on a 7500 Fast Real-Time PCR System (ABI, Los Angeles, CA, USA) with THUNDERBIRD^®^ Next SYBR^®^ qPCR Mix (TOYOBO, Osaka, Japan). The construction of the qPCR system and the amplification conditions were carried out according to the instructions of the THUNDERBIRD^®^ Next SYBR^®^ qPCR Mix. The relative mRNA expression levels were calculated using the 2^−ΔΔCt^ method. Statistical analysis of the relative mRNA expression level was performed using one-way ANOVA with SPSS 26.0 (v26) software, and the differences were considered significant when the *p* value < 0.05 (*) and extremely significant when the *p* value < 0.01 (**).

## 5. Conclusions

Notably, through a comprehensive investigation of the expression profiles of HSP70 genes under abiotic and biotic stresses, *hsp70* (*hspa1a*), *hspa1b*, and *hspa5* showed significant responses to all five diverse stresses, heat, salinity, and *E. scophthalmi*, *V. anguillarum*, and *Megalocytivirus* infection stresses, indicating their potential roles in comprehensive anti-stress responses. To date, most of the previous studies on stress in teleosts have focused only on a single stress condition. In contrast, comprehensive study on multiple stress conditions has only been reported for the spotted seabass, which showed that *hsp70.2* was involved in the response to various environmental stresses (heat, alkalinity, and hypoxia stress) [18]. To further validate the functions of these three candidate comprehensive anti-stress HSP70 genes (*hsp70* (*hspa1a*), *hspa1b*, and *hspa5*), we conducted heat stress and *E. tarda* infection stress experiments and performed qPCR analysis. The results showed that the expression of *hsp70* (*hspa1a*), *hspa1b*, and *hspa5* was indeed significantly or extremely significant upregulated after heat and *E. tarda*-infection stresses, indicating the crucial roles of these genes in response to biotic and abiotic stresses.

## Figures and Tables

**Figure 1 ijms-24-06025-f001:**
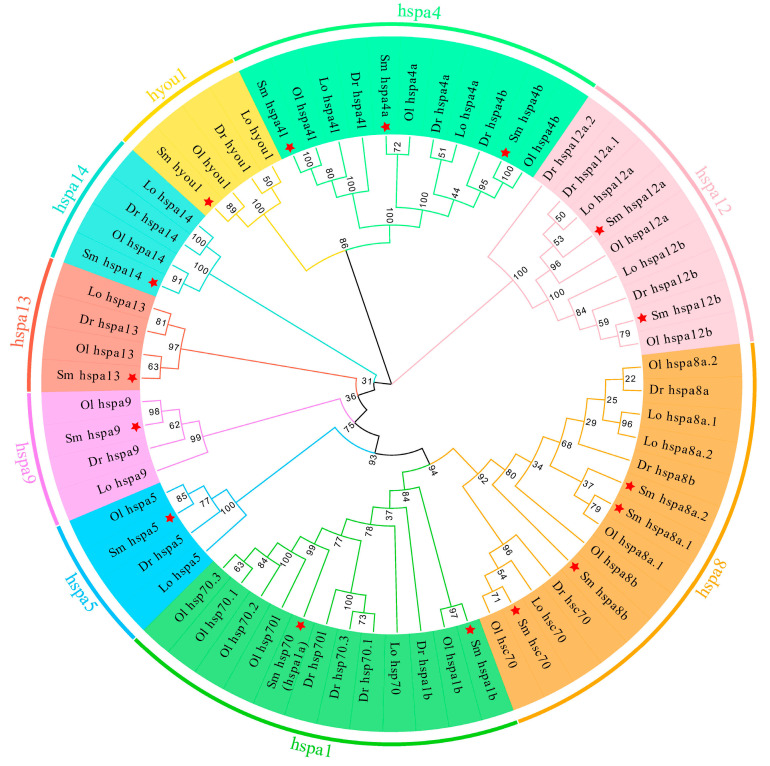
Phylogenetic analysis of HSP70 genes in four representative teleost species. A total of 66 HSP70 protein sequences from 4 teleost species were used to create the NJ tree using MEGA 11.0 with JTT model and 1000 bootstrap replications. Nine different groups were labeled as hspa1, 4, 5, 8, 9, 12, 13, 14, and hyou1 and distinguished by diverse colors. HSP70 genes of *S. maximus* were marked with a red asterisk. The abbreviations: Sm, *S. maximus*; Dr, *Danio rerio*; Ol, *Oryzias latipes*; Lo, *Lepisosteus oculatus*.

**Figure 2 ijms-24-06025-f002:**
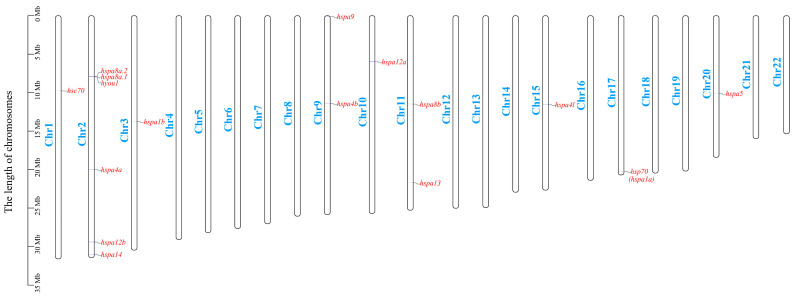
Chromosomal distribution of HSP70 genes in *S. maximus*. The HSP70 genes and chromosomes are highlighted in red and blue, respectively.

**Figure 3 ijms-24-06025-f003:**
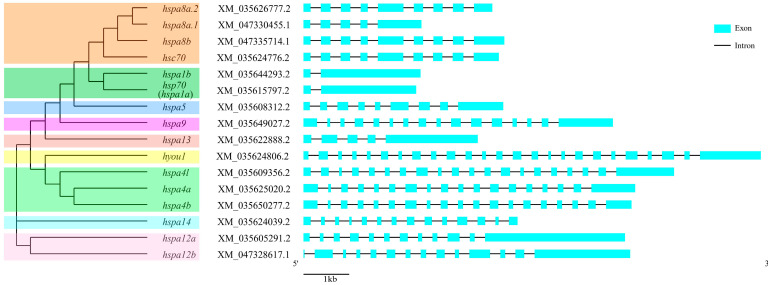
The combination of phylogenetic relationship and exon–intron structure. The left of the graph shows the phylogenetic relationships of *S. maximu* HSP70 genes. The right of the graph shows exon–intron structure and the length of introns of each HSP70 genes are displayed in the same length.

**Figure 4 ijms-24-06025-f004:**
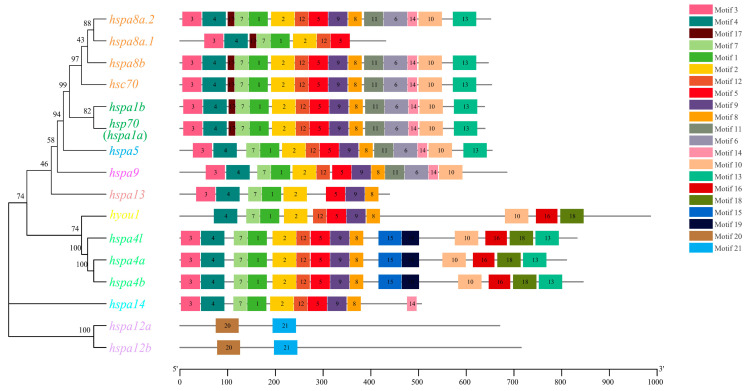
The combination of phylogenetic relationship and conserved motifs. The left of the graph shows the phylogenetic relationships of *S. maximu* HSP70 genes. The right of the graph shows the distribution of conserved motifs and each colored rectangular box represents a motif.

**Figure 5 ijms-24-06025-f005:**
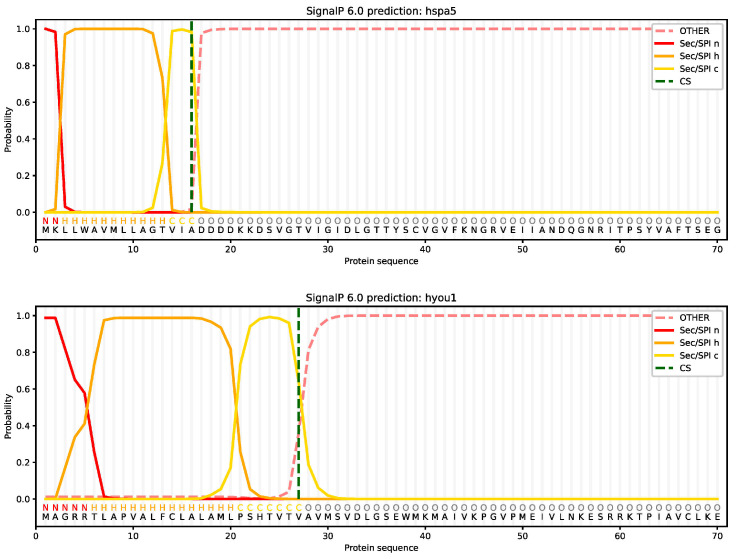
Signal peptide prediction analysis of the HSP70 proteins (*hspa5* and *hyou1*).

**Figure 6 ijms-24-06025-f006:**
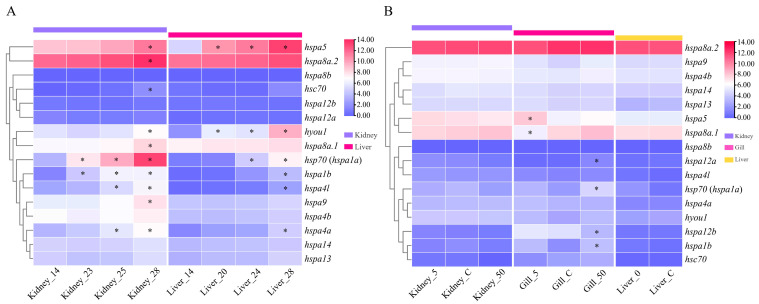
Expression patterns of HSP70 genes under abiotic stresses in *S. maximus*. (**A**) Expression patterns of HSP70 genes in kidney and liver tissues under heat stress. Control group: 14, 14 °C; 20, 23, 24, 25, and 28 represent 20 °C, 23 °C, 24 °C, 25 °C, and 28 °C heat stress groups, respectively. (**B**) Expression patterns of HSP70 genes in kidney, gill, and liver tissues under salinity stress. C: 30‰ (natural seawater); 50: 50‰; 5: 5‰; 0: 0‰ (fresh water). Cells with different colors correspond to different expression levels, which were normalized into log_2_(TPM + 1). * indicates differentially expressed (or significantly up- or downregulated) HSP70 genes with |log_2_fold change (FC)| > 1 and false discovery rate (FDR) < 0.01.

**Figure 7 ijms-24-06025-f007:**
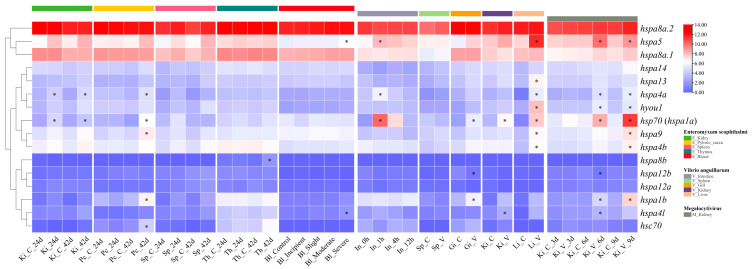
Heat map of HSP70 genes expression in *S. maximus* under pathogens (*E. scophthalmi, V. anguillarum,* Megalocytivirus) infection. Ki, Pc, Sp, Th, Bl, In, Gi, and Li represent kidney, pyloric caeca, spleen, thymus, blood, intestine, gill, and liver, respectively. C represents control group, V represents *V. anguillarum* infection, M represents Megalocytivirus infection. Cells with different colors correspond to different expression levels, which were normalized into log_2_(TPM + 1). * indicates the differentially expressed (or significantly up- or downregulated) HSP70 genes with |log_2_FC| > 1 and FDR < 0.01.

**Figure 8 ijms-24-06025-f008:**
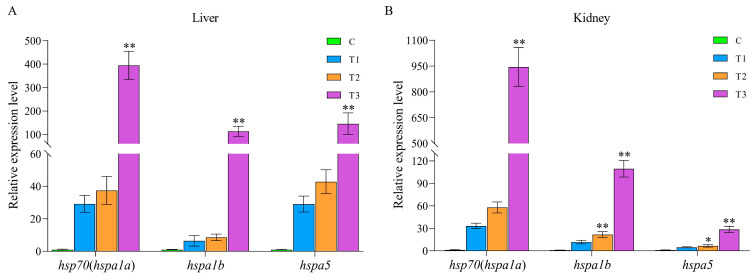
The validation of expression patterns of *hsp70* (*hspa1a*), *hspa1b*, and *hspa5* in liver (**A**) and kidney (**B**) tissues at 24 h after heat stress. The mRNA expression levels were determined by qPCR analysis using the 2^−ΔΔCt^ method. C, T1, T2, and T3 represent 18 °C, 22 °C, 26 °C, and 30 °C heat treatment groups, respectively. * and ** indicate the significant differences at *p* < 0.05 and *p* < 0.01 between the control (C) and heat treatment groups (T1, T2, and T3), respectively.

**Figure 9 ijms-24-06025-f009:**
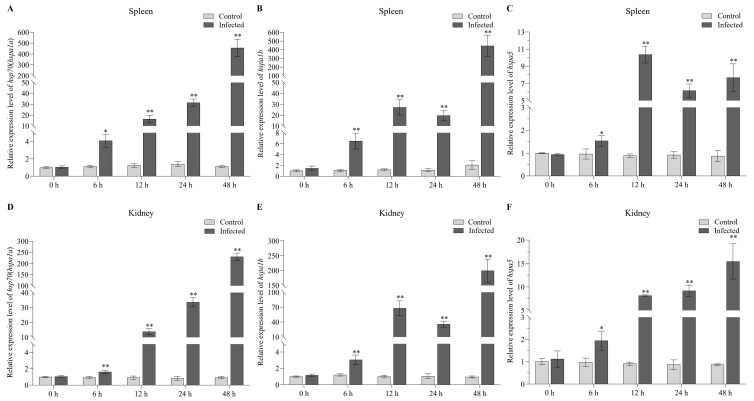
The validation of expression patterns of *hsp70* (*hspa1a*), *hspa1b*, and *hspa5* in spleen (**A**–**C**) and kidney (**D**–**F**) tissues at 0, 6, 12, 24, and 48 h post-infection with *E. tarda*. The mRNA expression levels were determined by qPCR analysis using the 2^−ΔΔCt^ method. * and ** indicate the significant differences at *p* < 0.05 and *p* < 0.01 between the control and infected groups at the same time, respectively.

**Table 1 ijms-24-06025-t001:** Summary of sequence characteristics of HSP70 genes in *S. maximus*.

Group	Gene Name	Chr	Gene Length (bp)	Numbers of Amino Acids	MW (Da)	pI	Domain	Domain Location (aa)
hspa1	*hspa1b*	Chr3	2780	638	70,307.59	5.42	HSPA1-2_6-8-like_NBD	8-383
	*hsp70* (*hspa1a*)	Chr17	2464	639	70,233.30	5.44	HSPA1-2_6-8-like_NBD	8-383
hspa4	*hspa4a*	Chr2	14,103	810	90,727.11	5.3	HSPA4_NBD	2-384
	*hspa4b*	Chr9	8098	845	94,849.45	5.04	HSPA4_NBD	2-384
	*hspa4l*	Chr15	8780	832	93,131.57	5.35	HSPA4_NBD	2-384
hspa5	*hspa5*	Chr20	3922	654	72,220.73	5.0	HSPA5-like_NBD	26-401
hspa8	*hspa8a.1*	Chr2	2990	431	48,133.63	6.87	HSPA1-2_6-8-like_NBD	52-367
	*hspa8a.2*	Chr2	4781	651	71,225.32	5.26	HSPA1-2_6-8-like_NBD	6-381
	*hspa8b*	Chr11	4378	646	70,684.97	5.36	HSPA1-2_6-8-like_NBD	6-381
	*hsc70*	Chr1	4665	653	71,324.65	5.27	HSPA1-2_6-8-like_NBD	6-381
hspa9	*hspa9*	Chr9	8965	685	74,100.10	6.09	HSPA9-like_NBD	52-428
hspa12	*hspa12a*	Chr10	28,355	670	74,993.54	6.57	HSPA12A_like_NBD	54-520
	*hspa12b*	Chr2	15,095	715	79,467.95	8.48	HSPA12B_like_NBD	57-554
hspa13	*hspa13*	Chr11	4707	439	47,641.75	5.91	HSPA13-like_NBD	23-431
hspa14	*hspa14*	Chr2	8663	506	54,385.87	5.99	HSPA14-like_NBD	2-376
hyou1	*hyou1*	Chr2	18,128	986	110,913.87	5.01	HYOU1-like_NBD	30-416

**Table 2 ijms-24-06025-t002:** *Ka*, *Ks*, and *Ka*/*Ks* ratios of duplicated HSP70 gene pairs.

Gene Pair	*Ka*	*Ks*	*Ka*/*Ks* Ratio
*hspa4a*-*hspa4b*	0.141613	1.856839	0.076266
*hspa4a*-*hspa4l*	0.251061	2.949802	0.085111
*hspa4b*-*hspa4l*	0.266541	3.017919	0.08832
*hspa8a.1*-*hspa8b*	0.14824	1.739731	0.085209
*hspa8a.2*-*hspa8b*	0.061431	1.285885	0.047773
*hspa8a.1*-*hspa8a.2*	0.120179	0.469765	0.255827
*hspa1b*-*hsp70*	0.080952	1.274974	0.063493
*hspa12a*-*hspa12b*	0.285699	1.211086	0.235903

**Table 3 ijms-24-06025-t003:** The secondary structure prediction and subcellular location prediction of HSP70 proteins in *S. maximus*.

Gene Name	Alpha Helix	Extended Strand	Beta Turn	Random Coil	Subcellular Location
*hspa4b*	43.55%	13.61%	3.31%	39.53%	cytosol
*hspa4a*	44.32%	14.20%	3.70%	37.78%	cytosol
*hspa4l*	43.87%	13.82%	3.73%	38.58%	cytosol
*hyou1*	50.51%	12.37%	3.75%	33.37%	cytosol
*hspa12a*	32.39%	19.10%	4.03%	44.48%	cytosol
*hspa8a.1*	41.76%	20.19%	5.10%	32.95%	extracellular
*hspa12b*	32.73%	19.02%	5.31%	42.94%	cytosol
*hspa13*	42.60%	21.18%	6.61%	29.61%	cytosol
*hsp70 (hspa1a)*	41.78%	18.94%	6.73%	32.55%	cytosol
*hspa8a.2*	41.32%	18.13%	6.76%	33.79%	cytosol
*hspa9*	41.61%	19.56%	6.86%	31.97%	mitochondrial
*hspa14*	37.55%	23.72%	6.92%	31.82%	cytosol
*hsc70*	41.50%	18.22%	7.04%	33.23%	nuclear
*hspa5*	44.04%	18.35%	7.19%	30.43%	extracellular
*hspa8b*	42.41%	17.96%	7.28%	32.35%	cytosol
*hspa1b*	43.10%	18.50%	7.37%	31.03%	endoplasmic reticulum

## Data Availability

The datasets analyzed for this study can be found in NCBI: https://www.ncbi.nlm.nih.gov/ (accessed on 7 March 2022). The accession numbers can be found below: SRP152627, SRP273870, SRP277001, SRP238143, SRP153594, SRP308109, SRP255305, SRP065375, SRP050607, SRP191266, SRP336094, SRP335896, SRP320422, SRP319434, and SRP347383.

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
