# Peer review of "Genome-Wide Identification, Molecular Characterization, and Involvement in Response to Abiotic and Biotic Stresses of the HSP70 Gene Family in Turbot (Scophthalmus maximus)"

_ijms, 2023, doi:10.3390/ijms24076025_

Round 1

Reviewer 1 Report

This is a voluminous and comprehensive study on genome-wide identification and molecular characterization of the heat shock proteins 70 (HSP70) in turbot. The study is well performed, the materials and methods is mostly detailed and the results are well presented.

Generally, abbreviations should be written in full the first time they are mentioned. Figure and Table legends should be written uniformly.

Materials and methods: 4.7 Biotic stress: Culture of pathogens: Should be written more detailed or references given.

Author Response

1. This is a voluminous and comprehensive study on genome-wide identification and molecular characterization of the heat shock proteins 70 (HSP70) in turbot. The study is well performed, the materials and methods is mostly detailed and the results are well presented.

Response:

Thank you for your comments and suggestions on our paper in your busy schedules. Thank you for your recognition and high evaluation of the results in our study.

2. Generally, abbreviations should be written in full the first time they are mentioned. Figure and Table legends should be written uniformly.

Response:

We have corrected and written the abbreviations in full the first time they are mentioned.

We have corrected the Figure and Table legends uniformly except for Figure5, because Figure 5 was generated automatically by the software and could not be edited.

3. Materials and methods: 4.7 Biotic stress: Culture of pathogens: Should be written more detailed or references given.

Response:

All of the biotic stress-related RNA-seq datasets used in 4.7 were retrieved from NCBI SRA database and references to each of these datasets were cited in the manuscript, so the detailed culture of pathogens could be found in the cited references. In our Edwardsiella tarda challenge experiment, E. tarda was cultured in lysogeny broth (LB) overnight for 24h at 28 ℃ using a shaker incubator, and we have added them in 4.9 of the manuscript.

Reviewer 2 Report

No comment. 

Author Response

Thank you for your review on our manuscript in your busy schedule.

Reviewer 3 Report

The authors provide new insights into the biological function of HSP70 in turbot adapting to various environmental stresses, but also contribute to the development of molecular selective breeding for comprehensive stress-resistance in turbot. I consider this manuscript makes a valuable contribution to the subject addressed. In general, the manuscript is well written, providing a concise discussion, I recommend the publication.  

Author Response

Thank you for your review on our manuscript in your busy schedule and thank you for your recognition and high evaluation of the results in our study.

Reviewer 4 Report

Dear Editors,

Dear Authors,

The manuscript entitled: “Genome-wide identification, molecular characterization, and involvement in response to abiotic and biotic stresses of HSP70 gene family in turbot (Scophthalmus maximus)” represents interesting study that aims to identify and characterize HSP70 gene family in turbot. Moreover, expression profiles of identified SHP70 genes under different abiotic and biotic stresses were determined in the species to check their functions. Because turbot is one of the most important marine species cultured in aquaculture and the presented results lay foundation for the future development farming practices aimed at the improvement of the species’ production effectiveness, the reviewed study deserves for publication. The applied means are appropriate to attain the specified by Authors aims and the substantive content of manuscript is correct. The manuscript requires, however,  requires some improvements, that include language presentation (especially abstract and introduction) and explanation some unclear terms. Conclusion chapter says the same like abstract and needs to be rewritten. I encourage the Authors to send the manuscript to the native speaker for corrections – some the most evident fixes I have put by myself.

In conclusion, I recommend the reviewed manuscript for publication in the IJMS periodical after major revision. All remarks, questions and fixes were placed in the attached pdf file (yellow highlights contain fixes and sentence suggestions, while red highlights contain comments and questions).

Thank you for another interesting manuscript that I could review!

Author Response

1. The manuscript entitled: “Genome-wide identification, molecular characterization, and involvement in response to abiotic and biotic stresses of HSP70 gene family in turbot (Scophthalmus maximus)” represents interesting study that aims to identify and characterize HSP70 gene family in turbot. Moreover, expression profiles of identified SHP70 genes under different abiotic and biotic stresses were determined in the species to check their functions. Because turbot is one of the most important marine species cultured in aquaculture and the presented results lay foundation for the future development farming practices aimed at the improvement of the species’ production effectiveness, the reviewed study deserves for publication. The applied means are appropriate to attain the specified by Authors aims and the substantive content of manuscript is correct.

Response:

Thank you for your review on our manuscript in your busy schedule and thank you for your recognition and high evaluation of the results in our study.

2. The manuscript requires, however, some improvements, that include language presentation (especially abstract and introduction) and explanation some unclear terms. Conclusion chapter says the same like abstract and needs to be rewritten. I encourage the Authors to send the manuscript to the native speaker for corrections – some the most evident fixes I have put by myself. In conclusion, I recommend the reviewed manuscript for publication in the IJMS periodical after major revision. All remarks, questions and fixes were placed in the attached pdf file (yellow highlights contain fixes and sentence suggestions, while red highlights contain comments and questions).

Response:

Thank you very much for your comments, suggestions and careful fixes on our paper in your busy schedules. We have studied your comments carefully and the conclusion chapter has been rewritten. In addition, we have sent the manuscript to the native speaker for revision and we have corrected them in the manuscript. The details modifications were marked up using the “Track Changes” function in the manuscript. Please see the resubmitted manuscript.

3. For those that I have not modified according to your comments, I would like to make a few points as following:

3.1. Notes on correction to the abstract

Response:

The abstract should be a total of about 200 words maximum, so we have not corrected exactly as your suggestion and fixes.

3.2. The picture is quite low quality.

Response:

Thank you for your suggestion. The pictures qualities were all 300 ppi in manuscript in Word format, but when we converted them to PDF format, the picture quality degraded a lot. In addition, we also submitted the vector images separately when we submitted the manuscript.

3.3. Please fix “lengths of introns of each HSP70 244 genes were displayed in the same length” in Figure 3 Caption.

Response:

Thank you for your suggestion. Some introns in many HSP70 gene members are very long and here we focus more on the number of introns and exons, so all of the introns are shown in the same length for better presentation in Figure 3 which was constructed using Gene Structure Display Server 2.0 software with the parameter “same length for introns”.

3.4. There is no control group on Figure 6A.

Response:

In Figure 6A, 14 represent 14℃ seawater group, and it is the control group.

Round 2

Reviewer 4 Report

Dear Editors,

Dear Authors,

The manuscript entitled: “Genome-wide identification, molecular characterization, and involvement in response to abiotic and biotic stresses of HSP70 gene family in turbot (Scophthalmus maximus)” has been significantly improved. Language presentation still requires style and grammar improvements – some fixes I have put by myself (attached file).

Good Job!

Author Response

Dear Reviewer,

Thank you for your recognition for our improvements and modifications in our resubmitted manuscript.

Thank you again for your fixes on our article patiently. We have corrected them according to your suggestion. Please see the resubmitted manuscript. 

Best regards.